# Nursing Students’ Experiences in School-Based Mental Health Promotion: A Qualitative Study in Chile

**DOI:** 10.3390/nursrep15120427

**Published:** 2025-11-29

**Authors:** Carmen Gloria Tapia Mercado, María Paz Sánchez-Sepúlveda, Daniela Solange Eichele Carrillo, Nolvia Francisca Muñoz Cárcamo, María Fernanda Lausen Correa, Karina Alejandra Osorio Vera, Maria Kappes

**Affiliations:** Facultad de Ciencias para el Cuidado de la Salud, Escuela de Enfermería, Universidad San Sebastián, Puerto Montt 5480000, Chile; carmen.tapia@uss.cl (C.G.T.M.); daniela.eichele@uss.cl (D.S.E.C.); nolvia.munoz@uss.cl (N.F.M.C.); maria.lausen@uss.cl (M.F.L.C.); karina.osorio@uss.cl (K.A.O.V.); maria.kappes@uss.cl (M.K.)

**Keywords:** school mental health services, students, nursing, health promotion, community of practice

## Abstract

**Background:** Mental health problems are increasingly common among children and adolescents, making schools a key setting for health promotion. Nurses can play a central role in prevention and support, but in Chile, the role of the school nurse has not yet been formally established. Understanding nursing students’ experiences in school-based mental health promotion can inform curriculum development and strengthen professional identity. **Methods**: A qualitative study with a phenomenological approach was conducted with third-year nursing students enrolled in a mental health course (N = 64). Data was collected through six individual interviews and one focus group, transcribed verbatim and analyzed using ATLAS.ti 25.0.1^®^. To ensure rigor, the study was guided by the Consolidated Criteria for Reporting Qualitative Research (COREQ), and trustworthiness was ensured following Lincoln and Guba’s criteria. **Results**: Five main categories emerged: nursing’s role in mental health promotion; perceptions of health promotion as prevention; use of the educational process, with difficulties in formulating objectives; perceived self-efficacy, marked by initial uncertainty in working with children; and experiences in implementing projects, including reflections on the professional role, mixed feelings, facilitators, barriers, and coping strategies. Facilitators included faculty support, teacher collaboration, and group cohesion, while barriers were related to limited experience, challenges in managing children, and external conditions such as noise and unsuitable classroom conditions. **Conclusions**: School-based practicums in mental health promotion are valuable opportunities to integrate theory and practice, strengthen professional identity, and develop communication. Strengthening undergraduate curricula with systematic training in these areas is essential for preparing nurses for their role in school and community health. Integrating these experiences into clinical and assistive practice can enhance early detection, interprofessional collaboration, and the promotion of healthier school environments.

## 1. Introduction

Chronic diseases, including mental health disorders, are on the rise worldwide, increasing the complexity of their management. Many risk behaviors begin in preadolescence and adolescence—periods that should be characterized by growth but are often affected by factors that compromise healthy development. Currently, one in seven adolescents between 10 and 19 years of age lives with a mental health problem, mainly depression, anxiety, or behavioral disorders, which are leading causes of morbidity and disability [1]. Additionally, they are more likely to be victims or perpetrators of bullying, reinforcing the need for early detection tools and effective interventions [2].

According to the International Council of Nursing, the discipline is an autonomous, collaborative practice aimed at comprehensive care and health promotion throughout the life cycle. Its sustained development in clinical practice and research has positioned nursing as a cornerstone of preventive strategies [3]. In school settings, nurses play a key role in identifying early difficulties, fostering trust, and strengthening healthy environments that support children and adolescents [3,4]. This requires competencies such as effective communication, confidentiality, and the ability to establish meaningful bonds [5], and poses challenges for training programs that must integrate clinical, communicational, and pedagogical skills [6].

Strengthening the nursing role is essential for managing chronic conditions and responding to the growing demands of health systems [7]. The COVID-19 pandemic highlighted nursing’s leadership in knowledge management, health education, and care delivery [8], as well as its capacity to enhance articulation between education and health sectors [9,10,11]. This includes the implementation of evaluation systems and digital tools that support training, promote healthy habits, and facilitate student monitoring [9,12,13]. These developments reveal the need for curricula capable of preparing graduates for dynamic epidemiological and social contexts [14]. In Chile, limited access to specialized mental health services for children and adolescents increases pressure on schools and underscores the need to reinforce the connection between the education and health sectors [15,16].

The World Health Organization defines health promotion as a set of actions aimed at strengthening individual and community capacities [16], while the Pan American Health Organization recognizes schools as strategic spaces for promoting healthy behaviors across the life course [16]. Within this framework, school nurses are essential for early identification of mental health problems, supporting academic engagement, and promoting student well-being [3,4,17,18].

Evidence indicates that trained facilitators can effectively promote self-awareness, self-care, and early detection of emerging problems [19]. Therefore, nursing education should adopt a lifelong learning approach [20], strengthening motivation, skill development, and self-esteem to reduce health inequalities [20,21]. However, not all nursing students prioritize this field, particularly mental health, and some show low interest in developing it professionally [22,23].

Effective mental health promotion requires specific competencies [24], innovative educational methods, and information technologies that enhance trust among learners and communities [24,25]. Students’ emotional well-being and life satisfaction influence academic adaptation and performance [26,27]. Although many graduates feel prepared to promote healthy lifestyles [20,28], barriers such as academic overload, unfavorable learning environments, anxiety, brief clinical practices, and insufficient faculty guidance persist [27,29,30].

Ozsaker et al. note that high stress levels during early clinical experiences require close faculty support, as promoting self-efficacy improves student performance, while low self-efficacy increases stress and disengagement [31]. Positive clinical environments and supportive relationships strengthen the professional identity and commitment [32]. Health promotion practices help consolidate competencies, integrate theory and practice, and build a solid professional identity [33]. Likewise, user-centered education facilitates reflective learning, empathy, and transformative practice [34].

From this perspective, and considering the advanced level of students’ training, it was hypothesized that clinical experiences in school-based mental health promotion would contribute to developing a more comprehensive and humanized understanding of mental health care. Accordingly, this qualitative study aimed to explore the experiences of nursing students during mental health promotion activities in school communities, identifying the facilitators and barriers that influenced their learning and professional role construction.

## 2. Materials and Methods

### 2.1. Objectives

Primary objective: “To explore the experiences of nursing students as they carry out mental health promotion activities within school communities”.

Secondary objectives:To describe nursing students’ perceptions of mental health promotion in school settings.To examine how students understand and apply the educational process during mental health promotion activities.To explore nursing students’ perceived self-efficacy when working with school-aged children.To identify the facilitators and barriers that shaped the implementation of mental health promotion activities.

### 2.2. Hypotheses

Given that this is a qualitative study, a research assumption was formulated.

Nursing students perceive themselves as capable of effectively delivering mental health education to school populations through strategies grounded in the educational process.

### 2.3. Study Design

A qualitative study was developed with a phenomenological approach, aimed at understanding the experiences lived by the participants. The research team was made up of nursing professors with doctoral and master’s degrees, and extensive experience in qualitative research. All had previous experience in studies related to nursing and mental health education, as well as training in interviewing techniques and conducting focus groups. Non-participant observation and semi-structured interviews were the main data collection strategies. The observations were used exclusively to provide context to the study, while the core techniques included individual interviews and a focus group. To ensure methodological rigor, COREQ guidelineswere considered in the design, data collection, and presentation of results [35], and are reported in Appendix A.

### 2.4. Participants and Setting

The study population consisted of third-year nursing students enrolled in the Mental Health course (N = 64) between April and June 2024. Within the framework of the community engagement project “Respectful Coexistence in Schools” (ID 3345), students carried out educational activities in both public and private schools. Nursing students were randomly distributed into 11 groups, each comprising six members.

### 2.5. Inclusion and Exclusion Criteria

Inclusion criteria: (1) third-year nursing students enrolled in the Mental Health Nursing course; (2) willingness to participate by signing an informed consent form after receiving prior information.

Exclusion criteria: (1) students who have failed their clinical experience.

### 2.6. Recruitment

A non-probabilistic convenience sample was employed, defined by the criterion of theoretical saturation. Data collection was considered complete once no new concepts emerged. The entire class was invited to participate (N = 64), of whom twelve agreed. Among these twelve, six students took part in individual interviews, reaching saturation by the fifth interview. An additional interview was conducted, along with a focus group, to enable methodological triangulation.

### 2.7. Sampling and Saturation

The sample was obtained through voluntary participation, resulting in six individual interviews and one focus group, which included six participants. Data saturation was reached during the fifth interview, when no new codes or concepts emerged. This was confirmed through iterative coding cycles and peer debriefing among the research team. For the focus group, participants who had agreed to take part in the study were randomly selected. This process led to the allocation of six individuals to the focus group and six to the individual interviews.

### 2.8. Data Collection

The interviews and the focus group were conducted by researchers, virtually through Microsoft Teams^®^ software version 2407. Each session lasted between 50 and 60 min. Field notes were taken during and after the interviews to capture nonverbal cues and contextual aspects. The researcher had no previous direct relationship with the participating students and did not participate in their academic evaluation. In addition, he was not an expert in school nursing, which minimized preconceptions about the topic and allowed participants to freely construct their own perspectives. All sessions were audio recorded with the consent of the participants and transcribed verbatim. Data collection was considered complete when no new concepts emerged and no repeat interviews were required. Transcripts were then returned to participants for review and validation to ensure the accuracy and fidelity of the accounts. After this process was completed, the audio files were permanently deleted to safeguard confidentiality.

### 2.9. Data Analysis

Data was analyzed using ATLAS.ti 25.0.1^®^, following three steps: (a) open coding to generate categories; (b) axial coding to establish relationships between categories and subcategories; and (c) selective coding, through which theoretical saturation was reached, and an interpretive narrative was developed. To enhance transparency in data analysis, we specify the following procedures. Coding was conducted collaboratively by two researchers. Initially, the lead researcher performed open coding using ATLAS.ti, highlighting significant verbatim excerpts and assigning preliminary codes. Subsequently, a shared codebook in Excel^®^ format was developed and iteratively refined across two cycles of axial and selective coding. Each iteration involved peer review and discussion with the other researchers to resolve discrepancies; disagreements were addressed through consensus after joint examination of the data. Examples of codebook evolution included merging overlapping codes (e.g., lack of confidence and self-efficacy concerns) and creating higher-order categories such as coping strategies. Intercoder agreement was not formally quantified, as the study adopted a consensus-based approach consistent with qualitative paradigms.

### 2.10. Methodological Rigor

Trustworthiness was ensured according to Lincoln and Guba’s criteria: credibility was ensured through regular peer debriefing sessions (biweekly), where coding decisions and thematic interpretations were critically examined. Transferability was supported by providing thick descriptions of context and participant quotations. Confirmability was addressed through original transcripts and analytical memos [36].

### 2.11. Ethical Considerations

The study was conducted in accordance with the ethical principles of research involving human subjects (Declaration of Helsinki). Ethical approval was obtained from the Research Ethics Committee under approval number (79/24). All participants provided electronic informed consent, ensuring voluntariness, confidentiality, and the right to withdraw at any time.

## 3. Results

Out of a total of twelve nursing students who agreed to participate in the study, six individual interviews and a focus group, composed of the remaining six students, were analyzed, which allowed us to triangulate perspectives and deepen the interpretation of the results. The analysis yielded five general categories with associated codes. Figure 1 shows the relationships between the categories, while Table 1 presents textual illustrative quotes that support the findings.

Five main categories emerged from the analysis. The first category, the role of nursing in mental health promotion, describes how students understood their function in the school setting, emphasizing education, support, and the creation of a safe environment for children. The second category, perceptions of promotion as prevention, reflects how participants viewed mental health promotion as an anticipatory strategy aimed at strengthening protective factors in schoolchildren. The third category, use of the educational process, highlights the relevance of pedagogical tools for developing activities, along with the challenges students faced when formulating learning objectives. The fourth category, perceived self-efficacy, captures the initial uncertainty experienced when interacting with children and how this influenced their confidence during the practicum. The fifth category, implementation of the project, brings together students’ reflections during the development of mental health activities, including mixed feelings, perceived facilitators, barriers encountered, and strategies they used to manage these situations.

Results: Thematic Categories, Descriptions, and Illustrative Quotes

Nursing’s role in mental health promotion

Participants underscored the importance of mental health and valued nursing as a coordinator and facilitator of tools to address difficulties. They recognized that, although they are not specialists in psychology or psychiatry, they can guide and accompany people through their processes.

“I see it more as someone who manages these kinds of things and can also help guide people… we can guide and accompany.” (I3)

2.Perceptions of mental health promotion

Students understood promotion to prevent the onset of illness and to offer support through educational interventions, distinguishing among different levels of action in health.

“What we’re aiming for is not to reach the development of diseases.” (I4)

3.Experience using the educational process

The educational process was valued as a useful tool, although they acknowledged difficulties, especially when formulating objectives with the appropriate taxonomy.

“Maybe it’s still a bit hard for us to choose the verbs for the objectives, because the activities feel like the easier part.” (I5)

4.Perceived self-efficacy

Before the interventions, students expressed uncertainty about their preparedness, particularly for working with children and managing groups. This reflected doubts about the actual effectiveness of their educational actions.

“I even questioned how we could do it so that they would really achieve meaningful learning.” (I6)

“I didn’t know—my main concern was how I was going to reach the children, how I could help them understand empathy.” (I2)

5.Experience implementing the educational project

Practice allowed students to reflect on their role in mental health promotion and to reframe their training experience. The focus group enriched this category by highlighting contrasts between frustration and satisfaction, as well as the construction of collective learning.

“As students, we have more freedom to express ideas.” (I5)

 (1)Subsequent feelings

Emotions ranged from frustration to satisfaction. Although some did not meet their initial expectations, they valued the experience as a learning opportunity which had an impact on schoolchildren.

“Honestly, I feel I didn’t achieve all the expectations I had at the beginning.” (I2)

“It was a good opportunity to build rapport with them and to provide something they can use.” (Focus group)

 (2)Facilitators

The facilitators identified included effective communication within the team, adequate organization, consistency in actions, leadership and support from both schoolteachers and teachers responsible for the groups of nursing students.

“What helped a lot was that we were consistent and held meetings regularly.” (I2)

 (3)Barriers

Barriers included limited experience in capturing children’s attention, poor communication among classmates, and external conditions such as small classrooms, cold temperatures, or ambient noise.

“There wasn’t good communication, so it was hard for us to organize ourselves ahead of time.” (Focus group)

 (4)Strategies to address challenges

Students proposed creative solutions such as dividing schoolchildren into small groups, incorporating playful activities, and relying on teachers to help manage discipline.

“The children’s teacher, who knew them better, was able to calm them down, which made things a bit easier for us.” (I2)

## 4. Discussion

Our findings show that students experienced both facilitators—such as faculty support and group cohesion—and barriers, including lack of experience and external conditions. These situations shaped the way they built their professional role. In Chile, this process takes place in a particular context: although the Chilean State, through the national health strategy for health objectives by 2030 [37], recognizes the importance of school health, it has not formally incorporated the school nurse into the educational system. This contrasts with countries such as Argentina [3], the United States [38], and Canada [39], where the school nurse is a key factor in mental health promotion and prevention. This difference limits learning opportunities and makes it necessary to create curricular and practical spaces that compensate for this absence [4,40].

Participants emphasized the importance of mental health and perceived nursing as a coordinating and facilitating agent, capable of guiding and supporting individuals despite not being mental health specialists. They understood health promotion primarily as disease prevention through educational interventions; however, their initial uncertainty regarding self-efficacy—particularly in group management and interaction with school populations—underscored the need for structured training. Practical experience, supported by schoolteachers, transformed these challenges into meaningful learning opportunities, fostering collective reflection and contributing to the development of a professional identity. These findings resonate with a qualitative exploratory study in Norway involving 284 school nurses [5], which showed that nurses lacked the knowledge and confidence to work with children and adolescents with mental health problems, reinforcing the need for enhanced education in this area [5,6,7].

These results also indicate that the construction of the nursing role emerged through the integration of theoretical knowledge with the realities of classroom environments. As students navigated these scenarios, their understanding of the nursing role expanded beyond clinical tasks toward a more holistic view that incorporated educational and community-oriented responsibilities. Similar international studies suggest that experiential learning strengthens communication skills and helps establish foundational competencies for working with school-aged populations.

In this context, undergraduate training becomes especially relevant. Previous literature has highlighted its importance for preparing nurses to deliver health education, including mental health [6,7,14]. Participants in this study also saw it as essential. Yet, they tended to describe health promotion mainly as a supportive or managerial task, where the professional provides or mobilizes resources. This view does not fully match what international agencies propose, where promotion also means proactive, population-level action to strengthen skills and environments [16]. This gap may reflect the fact that our participants were students facing their first clinical experience in mental health promotion. Additionally, the absence of school nurses in their practice settings may have reduced their perception of the importance of this role. Even so, their views shifted after completing the activities, demonstrating a growing understanding of the professional impact of health promotion.

A key element was self-efficacy. Students often doubted their ability to manage groups or teach children effectively, which made mental health promotion less attractive to some [22,23]. Similar experiences have been reported elsewhere, showing that more structure and discipline are needed in teaching and practice. Trained facilitators have shown positive effects on promotional interventions [19], and our participants’ experiences support this. They also identified gaps in communication, teamwork, and educational strategies, consistent with Pérez-Avilés [27]. Facing these challenges encouraged deeper reflection and helped them recognize the complexity of teaching in health settings, ultimately motivating them to redefine aspects of their emerging nursing role.

At the same time, both barriers and facilitators played a decisive role in learning. Barriers included communication difficulties among peers, which affected work organization, and contextual challenges within schools, which influenced their learning process, as reported by McTier, Phillips, and Duke [27]. Conversely, facilitators—particularly faculty support during practicum—gave students more confidence, a finding consistent with Moktan and Mehta [24]. Support from schoolteachers complemented academic guidance, creating a learning environment where group cohesion and creativity helped overcome obstacles such as lack of experience or noisy classrooms. These factors turned initial insecurities into meaningful opportunities for skill development.

From a clinical practice perspective, our findings show that students progressively applied core nursing competencies—such as communication, assessment, and leadership—during school-based mental health activities. This aligns with previous evidence [5,6,7,19,27] highlighting the importance of these skills in community and child–adolescent mental health care.

Overall, these findings show that clinical learning in health promotion is central to shaping professional identity and role ownership during undergraduate education. Guidance, mentorship, and respectful accompaniment from faculty are crucial for lasting learning. Therefore, curricula should integrate, particularly from the early stages, enhanced training in communication, teamwork, group management, and public speaking, which are all essential skills for mental health promotion—with the goal of fostering nursing as an autonomous and collaborative practice.

## 5. Limitations

This study has several limitations that should be acknowledged. First, the sample consisted of a small number of participants from a single private university in southern Chile, which limits the representativeness and transferability of the findings. Although data saturation was achieved through iterative analysis—consistent with phenomenological qualitative methodology—the results cannot be generalized to other educational settings. Future studies including students from multiple universities and public institutions would strengthen the breadth of perspectives.

Second, the use of a single focus group and a limited number of individual interviews restricted the depth of triangulation by techniques. Expanding the number of focus groups in future studies could enhance methodological robustness.

Third, the phenomenological qualitative design—while appropriate for exploring lived experiences—does not allow extrapolation or quantification of results.

Fourth, data were collected through virtual interviews and a virtual focus group, which may have reduced the ability to observe nonverbal communication cues, although field notes were used to mitigate this limitation.

Fifth, although researcher triangulation and participant validation were applied to enhance rigor, the possibility of interpretive bias cannot be entirely excluded.

Finally, the particular context of the Chilean educational system—where the role of the school nurse is not formally recognized—may have influenced students’ perceptions of their professional role and self-efficacy.

Despite these limitations, the study offers valuable insights into how school-based mental health promotion contributes to the construction of nursing students’ professional identity and highlights areas for curriculum development and training.

## 6. Conclusions

School-based mental health promotion practicums constitute essential formative experiences that enable nursing students to integrate theoretical knowledge with real-world practice and to strengthen core disciplinary competencies. In this study, students demonstrated progressive development of communication, leadership, and pedagogical skills, as well as increased clarity regarding their emerging professional identity. Faculty mentorship and collaboration with schoolteachers operated as key enabling conditions, whereas barriers such as limited prior experience and challenging school environments highlighted the need for more structured preparation before clinical engagement. These findings support the early and systematic incorporation of training in communication, group management, and educational strategies into undergraduate curricula—particularly in contexts where the school nurse is not formally recognized within the educational system. Such training may enhance students’ readiness for community and school-based mental health roles and improve the continuity between academic preparation and clinical practice. Future research should examine how repeated exposure to school-based mental health promotion influences the long-term consolidation of the nursing professional role throughout the academic trajectory.

## Figures and Tables

**Figure 1 nursrep-15-00427-f001:**
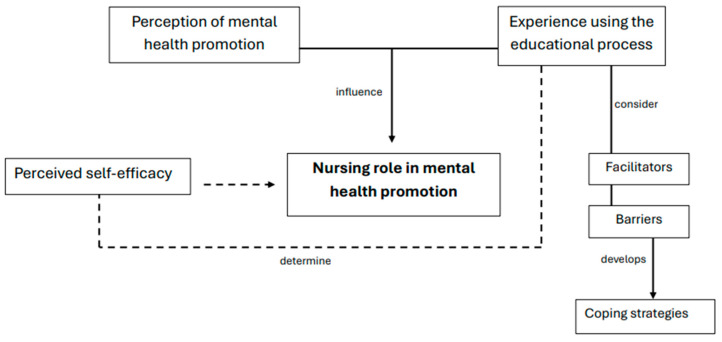
Relationship among categories.

**Table 1 nursrep-15-00427-t001:** Categories, subcategories, and verbatim quotations from nursing students on mental health promotion.

Category	Subcategory/Code	Example Quote
1. Nursing role in mental health promotion	Coordinating role and accompaniment	“…we can guide and accompany.” (I3)
2. Perception of mental health promotion	Prevention and differentiation of levels of intervention	“…we aim to prevent the development of illnesses.” (I4)
3. Experience using the educational process	Difficulty in formulating objectives	“…we still find it a bit difficult to choose the verbs for the objectives.” (I5)
4. Perceived self-efficacy	Uncertainty about working with children	“…my main concern was how to reach the children.” (I2)
5. Experience in implementing the educational project	Reflection on the professional role	“…as students we have more freedom to express ideas.” (I5)
5.1. Subsequent feelings	Frustration and satisfaction	“…I feel I did not meet all the expectations.” (I2)
5.2. Facilitators	Faculty support and group cohesion	“…what helped a lot was that we held regular meetings.” (I2)
5.3. Barriers	Lack of experience and external conditions	“…communication was not good, so organizing ourselves was hard.” (Focus group)
5.4. Coping strategies	Creative solutions and faculty support	“…the head teacher was able to calm them down and that made things a bit easier for us.” (I2)

## Data Availability

The data are available upon direct request to the authors.

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
