# Peer review of "Nursing Students’ Experiences in School-Based Mental Health Promotion: A Qualitative Study in Chile"

_nursrep, 2025, doi:10.3390/nursrep15120427_

Round 1
Reviewer 1 Report
Comments and Suggestions for Authors
Recommendation: Reconsider after major revision. The study is timely and within scope, but the manuscript needs substantial editing and methodological clarification before it is suitable for publication. Presentation issues include residual template text and a broken sentence in the Results, plus duplicated lines in Methods, which impede readability (e.g., MDPI placeholder text; repeated “transcripts were returned…”). Sampling and saturation also need a tighter rationale, given a course cohort of N=64 but only six interviews plus one focus group analyzed. Strengthen the Results with thicker description (more varied quotations with participant IDs per theme) and ensure Figure 1/Table 1 convey real analytic structure, not labels.
The topic is timely and suitable for the journal, but the manuscript requires major corrections in presentation and methodological transparency. (1) Presentation: The Results section still contains template/placeholder text (e.g., “This section may be divided…”) and a sentence that breaks mid-line (“…that can”). Remove all boilerplate, repair broken lines, and open Results with a concise thematic overview. In Methods, consolidate duplicated content (e.g., repeated statements about virtual interviews and returning transcripts to participants). Ensure the title and headings are free of awkward hyphenation and that any editorial placeholders in the front matter are either completed or removed. (2) Sampling and saturation: The course cohort is reported as N≈64, yet the analyzed data comprise six individual interviews plus one focus group (~6 participants). Provide an auditable rationale for sample size and stopping: describe the saturation criterion used, how redundancy was assessed (e.g., saturation grid or tracking), and clarify the selection process for the focus group (“randomly invited” → how implemented). (3) Analysis transparency: You reference ATLAS.ti and open/axial/selective coding, but it remains unclear who coded, whether coding was independent or collaborative, how disagreements were resolved, and how the codebook evolved across cycles. Add coder roles, a brief account of codebook iteration (with examples), and a note on any interceder agreement process (or the rationale for not using one). Move beyond generic trustworthiness claims; specify exactly how credibility/transferability/dependability/confirmability were enacted (e.g., peer debriefing cadence and outcomes, member-checking returns, audit-trail artifacts). (4) Coherence across sections: The design lists non-participant observation and personal narratives, but these sources are not clearly represented in Results. Either integrate their contributions (with identifiers) or state explicitly that they informed context only. (5) Language and references: Apply a light language edit to remove repetitions and improve flow. Balance institutional web sources with peer-reviewed literature aligned to school-based mental-health promotion and nurse education, and ensure all in-text citations appear in the reference list. In sum, the study’s contribution is promising, but publication readiness depends on removing template text, disambiguating sampling/saturation, fully documenting the analytic procedures, and strengthening the evidentiary depth and visualization of findings.
Author Response
Dear Reviewer 1, please find attached a compiled document containing the responses to all reviewers. Kind regards

Reviewer 2 Report
Comments and Suggestions for Authors
Dear Authors, Please find below my suggestions: The title would benefit from greater clarity. I suggest specifying more clearly the type of study conducted and the setting (e.g., "China"). The conclusions of the Abstract could benefit from an extension that considers implications for clinical and assistive practice. The keywords are somewhat redundant; I recommend selecting 4–5 terms that closely reflect the main topics of the study and explicitly including both the study design and setting (as also suggested for the title). Overall, the introduction is sufficiently structured, but it lacks a key component: a conceptual framing of the nurse’s role in managing and providing care within a clinical practice perspective, mainly in chronic care and Covid period. In this regard, and with the support of recent and appropriate literature, I suggest incorporating relevant recent topics such as Impact of School Nurse on Managing Pediatric Type 1 Diabetes with Technological Devices Support, School Nurses' Available Education to Manage Children with Asthma at Schools, and Engaging School Communities During COVID-19 that would certainly help to strengthen the rationale of the proposed work and open the discussion to an international and multi-dimensional perspective. The objectives (as I interpret them from lines 97–103) are not clearly stated and require careful revision. I suggest using the standard format: “The primary objectives were... while the secondary objectives were...” Also, include the specific research questions. Given their importance, I recommend creating a dedicated subsection such as "1.1 Aims", or placing them at the beginning of the Methods section. Still regarding the Methods, there is a lack of detail concerning the specific methodology applied and its internal validity (inclusion and exclusion criteria, recruitment process); please clarify these aspects. For the sake of transparency, the reporting checklist for the declared reporting tool is missing from the supplementary files—this should be added. The Results are certainly a strength of the study, but they require attention in terms of editing and presentation of figures and tables, which are currently not entirely clear. The Discussion (see also the earlier comment on the Introduction) lacks a real interpretation from a clinical and assistive practice perspective. I suggest creating a dedicated section to expand on the implications—drawing from both the introductory suggestions and the findings of the present study. This should be supported by updated and appropriate references. The Limitations deserve a dedicated section and should be significantly expanded, as in their current form they are overly simplistic and superficial. The Conclusions need to be revised and updated in accordance with the comments above, and the References must be strengthened to support those points. Any references older than ten years should be updated—unless they are strictly methodological or represent high-impact, evidence-based sources. Providing a point-by-point response to the comments will be essential to complete the process of international recognition of the work, particularly in terms of methodology and the clinical and assistive practice perspective.
Comments on the Quality of English LanguageNative review recommended
Author Response
Dear Reviewer 2, please find attached a compiled document containing the responses to all reviewers. Kind regards

Reviewer 3 Report
Comments and Suggestions for Authors
The topic of this research is very important.
But a much deeper theoretical background should be provided, and it should also be approached from an international perspective. And since the students are in their third year, their training to date should also be evaluated (based on what we expect and what we do not expect – precise research questions and hypotheses are needed, taking into account every possible aspect). Studies cannot be ignored when answering questions.
The sample is small, so it is very difficult to draw robust conclusions, even with precise analysis. Since the sample size cannot be changed, the analysis must be much more detailed, which requires detailed foundations (e.g., background, curicullum analysis).
In addition to what they have learned, it would also be worthwhile to explore their own attitudes, meaning that the research could have been supplemented with a questionnaire technique. In its current form, I consider this work to be the beginning of a very important research project. This is also worth emphasizing. I trust that the research will be continued.
The English could be improved to more clearly express the research.
Author Response
Dear Reviewer 3, please find attached a compiled document containing the responses to all reviewers. Kind regards

Reviewer 4 Report
Comments and Suggestions for Authors
I'm attaching a file.

That's ok.
Author Response
Dear Reviewer 4, please find attached a compiled document containing the responses to all reviewers. Kind regards

Reviewer 5 Report
Comments and Suggestions for Authors
Dear authors.
I appreciate the thorough work and the valuable insights it provides into nursing students’ experiences in school-based mental health promotion. The study addresses an important topic and offers meaningful contributions to the field. I offer some suggestions for improvement to help strengthen the manuscript further:
The wording could be improved by avoiding redundancies in the description of the interview platform, transcription and validation, as well as data saturation, in order to improve the clarity and fluidity of the text for the reader.
The methodology is robust, but specific examples of the application of Lincoln and Guba's criteria could be provided. In addition, the implicit limitations could be mentioned and what was done to minimise bias could be explained.
The results clearly describe the facilitators (teacher support, group cohesion) and barriers (lack of experience, external conditions), which is positive, but they are repeated several times. Ideas about the importance of training, facilitators and barriers are repeated several times, making the reading redundant.
It is also advisable to avoid redundancy in the discussion. Although the limitation of the single university and the qualitative design are mentioned, it could be expanded upon how this affects the interpretation of the findings. It concludes with implications for university education but does not clearly summarise the main contribution of the study on the construction of the professional role.
In the conclusions section the authors do a good job of summarising the importance of the practicum in school mental health for integrating theory and practice, highlighting key facilitators such as teacher support and collaboration with the school, and pointing out implications for university training, such as training in communication and teaching strategies. It also proposes future lines of research on the consolidation of the professional role. However, it has some weaknesses: it does not explicitly summarise the main contribution of the study, which is how the concrete experience of students influences the construction of their professional role. Likewise, the conclusions could avoid the repetition of ideas that are already mentioned several times in the results, such as the importance of teacher support.
Although the references are relevant and add value, the reference section needs to be reviewed according to the journal’s guidelines. Typographical errors and inconsistent DOI formats have been identified and need correction.
Yours faithfully.
The reviewer
Author Response
Dear Reviewer 5, please find attached a compiled document containing the responses to all reviewers. Kind regards

Round 2
Reviewer 2 Report
Comments and Suggestions for Authors
Dear Authors,
the refernces (7-9) adopted for support the validity of the topic extended according previuous comment "of the nurse’s role in managing and providing care within a clinical practice perspective, mainly in chronic care and Covid period" aren't of sufficient international level, as they come from non-indexed sources lacking an impact factor, and therefore cannot adequately support the contents of the manuscript. The other additions in this version are often outdated and, in any case, extremely generic. The objectives remain difficult to understand (especially the secondary ones) and are not consistent with the results obtained and the hypotheses proposed. The internal validity of the study is not supported by the extremely limited representativeness of the sample, which, moreover, is not even mentioned among the study’s limitations. Fundamental elements for this type of study are also missing, such as data saturation. The narrative synthesis of the collected data is poor and overly easly, preventing a real interpretation of the findings. The discussion remains lacking in a real interpretation from a clinical and care practice perspective and is not supported by scientific literature in the respective sections (lines 326–336, 358–368). The conclusions also require academic adjustments to achieve greater scientific validity.
Comments on the Quality of English LanguageNative review recommended
Author Response
Dear Reviewer 2,
Please find attached my responses to your comments.
Kind regards,
María Paz Sánchez-Sepúlveda

Reviewer 3 Report
Comments and Suggestions for Authors
The authors corrected what they could. But I still maintain that deeper analysis was needed. It seems that the authors did not want this.
Comments on the Quality of English LanguageThe English could be improved to more clearly express the research.
Author Response
Dear Reviewer 3,
Please find attached my responses to your comments.
Kind regards,
María Paz Sánchez-Sepúlveda

Reviewer 5 Report
Comments and Suggestions for Authors
Dear Authors.
The work shows remarkable progress. The conclusions are now much clearer and more focused, which really highlights the impact of the study. The methodology has also been refined, demonstrating a more robust and well-structured approach. It is evident that considerable effort has been devoted to improving the quality of the manuscript.
Although the wording has been improved, there are still redundant ideas in the introduction.
Redundancy in the mention of mental health promotion:
The topic of mental health and its promotion is mentioned several times in a similar way. For instance, how mental health promotion is essential and how mental health programmes contribute to early detection.
Redundancy regarding the role of nurses:
The importance of the role of nurses in the early identification of mental health problems, supporting academic performance and the overall development of students is repeated several times.
Repetition of the mention of the benefits of nurse intervention:
The benefits of nurse intervention, such as academic support and the promotion of healthy development, are mentioned repeatedly in several sections.
Redundant development of the relationship between mental health and education:
The connection between mental health, the promotion of healthy habits, and the improvement of the educational environment is presented several times in a similar way, which could be consolidated to avoid repetition. It is advisable to consolidate the ideas to optimise the structure of the theoretical framework and improve the flow of the text.
References:
The manuscript needs to be improved by standardizing all references to the journal's format. References 2, 5, 7, 30, 31, 32, 34, and 35 currently use the doi: format, while the rest follow the APA 7 https://doi.org/xxxxx format. Additionally, all references should be reviewed.
Regards.
The Reviewer.
Author Response
Dear Reviewer 5,
Please find attached my responses to your comments.
Kind regards,
María Paz Sánchez-Sepúlveda
